# Online Neural Sequence Detection with Hierarchical Dirichlet Point Process

**Weihan Li**[1,3]**, Yu Qi**[2,3,*]**, Gang Pan**[3,1,*]

[1]College of Computer Science and Technology, Zhejiang University, Hangzhou, China
[2]The Affiliated Mental Health Center & Hangzhou Seventh People's Hospital,
the MOE Frontier Science Center for Brain Science and Brain-machine Integration,
Zhejiang University School of Medicine, Hangzhou, China
[3]The State Key Lab of Brain-Machine Intelligence, Zhejiang University, Hangzhou, China
`weihanli@zju.edu.cn, qiyu@zju.edu.cn, gpan@zju.edu.cn`

## Abstract

Neural sequence detection plays a vital role in neuroscience research. Recent impressive works utilize convolutive nonnegative matrix factorization and Neyman-Scott process to solve this problem. However, they still face two limitations. Firstly, they accommodate the entire dataset into memory and perform iterative updates of multiple passes, which can be inefficient when the dataset is large or grows frequently. Secondly, they rely on the prior knowledge of the number of sequence types, which can be impractical with data when the future situation is unknown. To tackle these limitations, we propose a hierarchical Dirichlet point process model for efficient neural sequence detection. Instead of computing the entire data, our model can sequentially detect sequences in an online unsupervised manner with Particle filters. Besides, the Dirichlet prior enables our model to automatically introduce new sequence types on the fly as needed, thus avoiding specifying the number of types in advance. We manifest these advantages on synthetic data and neural recordings from songbird higher vocal center and rodent hippocampus.

## 1 Introduction

Neural sequences are a set of sequential firing neurons repeatedly occurring over time, and they play crucial roles in understanding the brain activities of working memory [6, 10], motor production [9], and memory replay [4]. However, in general, it's hard to directly observe neural sequences from high-dimensional neural recordings due to the unsorted neurons. Besides, methods that use linear dimensionality reduction of neural populations may be inappropriate to handle such sequences since they form a high-dimensional structure and cannot be efficiently summarized by a low-dimensional representation [15, 23]. Finally, the relationship between neural sequences and behavior is not always so regular and predictable [30]. Thus an unsupervised method to identify neural sequences is required.

While recent works [15, 20, 28] have made a great process in neural sequence detection, they have two drawbacks when it comes to the contexts of online learning. Firstly, they are ill-suited to accommodate new data or handle large datasets. Many real-world neural recordings involve many hours' or days' signals that sometimes may be hard to accommodate entirely in the memory. Furthermore, the inference algorithms of these works (e.g., coordinate descent, Gibbs sampling, etc.) maintain the entire data and perform iterative updates of multiple passes, which may be a waste of both computational and storage resources. Secondly, these works require the number of sequence types to be specified in advance, which is inappropriate when a future situation is unknown.

---

*Corresponding authors: Yu Qi and Gang Pan

36th Conference on Neural Information Processing Systems (NeurIPS 2022).

To address these problems, we propose a hierarchical Dirichlet point process (HDPP) with an efficient online inference algorithm. Briefly, we use a two-level structure in our approach. At a lower level, we model the set of neural sequences by a Dirichlet nonhomogeneous Poisson process: The observed spikes are considered as a mixture of neural sequences, and those spikes belonging to different sequences are modeled via different nonhomogeneous Poisson processes, for which the intensity of each process varies across neurons and times. At a higher level, we model the types of sequences by a Dirichlet Hawkes process[5]: The detected sequences are partitioned into an automatically inferred number of types. The temporal dynamics of these sequences are captured by several Hawkes processes [11], where each process's distinctive temporal relation will help our model to disambiguate different types of sequences with spatial overlapping neurons. Besides, the prior of each sequence type is given by a Dirichlet distribution, so either the number of sequences or sequence types could vary during inference. Note that our model has only one Dirichlet point process at the lower level, which differs from the classical hierarchical Dirichlet process (HDP) [25].

We derive a Particle filter [2, 21, 22] to learn the HDPP in an online manner: (1) learn the model with a single pass of the data; (2) make the decisions regularly without requiring future data [1]. In particular, we use a collection of particles to sequentially approximate the posterior, taking in one spike at a time to make the update. We also develop specific mechanisms to merge similar sequences and prune inaccurate ones during inference, thus further improving the performance. We evaluate the proposed model on synthetic data as well as real-world data from songbird higher vocal center [15] and rodent hippocampus [3, 7, 8]. Results show the proposed model can properly identify the neural sequences and be robust to noise in a single pass over the data.

The contributions of our work include: (1) We propose a novel hierarchical Dirichlet Point Process for neural sequence detection, and different from prior works [14, 15, 28], our model does not specify the number of sequence type in advance, while determines adaptively during inference. (2) Our model with an efficient online inference algorithm performs comparably to the state-of-the-art [28] with a significantly lower time cost. (3) We use a group of Hawkes processes to capture the temporal dynamics of different types of sequences, which enables our model to distinguish sequences with spatial overlapping neurons.

## 2   Related works

Previous studies about neural sequence detection can be divided into three classes: supervised, bottom-up unsupervised, and top-down unsupervised [32].

**Supervised methods.** Averaging spikes times over multiple trials is the simplest and most used supervised method to identify neural sequences. This approach requires simultaneous sensory cues or behavioral actions recording and uses a template matching procedure to discover exact timestamps of each sequence [12, 18, 19]. However, as previously mentioned, the relationship between neural activity and behavior is not always reliable [30]. Thus this approach may be useless under certain circumstances.

**Bottom-up unsupervised methods.** Bottom-up unsupervised methods [23, 24, 26] usually identify neural sequences via a variety of statistics between pairs of neurons (e.g., cross-correlations). These methods may be beneficial when involving small numbers of neurons [32], but they have high computational complexity and low statistical power when dealing with a large number of neurons.

**Top-down unsupervised methods.** Top-down unsupervised methods can be defined as a kind of method that discovers a sequence from the perspective of the whole population and is robust to noise at the level of individual neurons [32]. Maboudi et al. [14] identifies neural sequences by visualizing the transition matrix of Hidden Markov Models. Peter et al. [20] and Mackevicius et al. [15] use convolutive nonnegative matrix factorization (ConvNMF) with different regularizations to factorize the neural data into a couple of neural factors and temporal factors, where each neural factor encodes a type of neural sequence and each temporal factor indicates the times of sequences. Williams et al. [28] models neural sequences as a Neyman-Scott process (PP-Seq), a spatial-temporal point process originally designed to discover clusters in the galaxy. Each sequence in this model corresponds to a latent cause that generates spikes as the offspring events.s

## 3   Background

A basic probabilistic model of continuous-time event stream is temporal point process whose realization can be represented as an event sequences $\{x_i\}_1^N = \{t_1, t_2, ...\}$ with times of occurrence $t_i \in [0, T]$. A way to characterize temporal point processes is via intensity function $\lambda(t)$, for which $\lambda(t)$ equals the expected instantaneous rate of happening next event given the history. The functional form of $\lambda(t)$ varies among different phenomena of interest. In a *homogeneous* Poisson process (PP), $\lambda(t)$ is assumed to be a constant over time, while in a *nonhomogeneous* Poisson process, $\lambda(t)$ is time-varying and independent of the history.

**Hawkes process.** A Hawkes process (HP) [11] is a self-exciting nonhomogeneous Poisson process in which past events in history have influenced the intensity of current and future events. Such influences are positive but decay exponentially with time:

$$\lambda(t) = \lambda_0 + \alpha \sum_{t_i < t} \exp(-\delta(t - t_i)), \tag{1}$$

where $\lambda_0$ is the base intensity independent of the history, $\alpha$ is the kernel parameter, and $\delta$ is the decay rate of that influence. An important extension of the Hawkes process is to incorporate additional information about each event, for example, an associated event "label" $k_i$ with each event $x_i = (t_i, k_i)$. As a consequence, we can define a Hawkes process as a sum of independent Hawkes processes from different types based on the superposition principle: $\lambda(t) = \sum_{k=1}^{K} \lambda_k(t)$, where $\lambda_k(t)$ is the intensity function from $k$-th category.

**Dirichlet Hawkes process.** Dirichlet Hawkes process (DHP) [5, 16] is a method to naturally handle infinite mixture clusters with temporal dependency. As opposed to traditional parametric models, DHP allows the number of clusters to vary over time. It uses a Hawkes process to model the intensity of events (e.g., the times of neural sequences), while the Dirichlet process captures the diversity of event types (e.g., the clusters of neural sequences). Similar to the Dirichlet process, a DHP, typically denoted by $\text{DHP}(\lambda_0, \theta)$ is characterized by a concentration parameter $\lambda_0$ and a base distribution $\theta$. It uses the samples $\theta_{1:K}$ from base distribution $\theta$ and a collection of triggering kernel functions $\phi_{\theta_i}(t, t_i)$ as the parameters for modeling $K$ different event types. Then, the events $(t_i, \theta_i)$ from $\text{DHP}(\lambda_0, \theta)$ can be simulated as follows:

1. Sample $t_1$ from $\text{PP}(\lambda_0)$ and $\theta_1$ from $\theta$.

2. For $n > 1$, sample $t_n$ from $\text{HP}(\lambda_0 + \sum_{k=1}^{K} \lambda_k(t_n))$

   - Sample a new $\theta_{K+1}$ from $\theta$ with probability: $\lambda_0 / (\lambda_0 + \sum_{k=1}^{K} \lambda_k(t_n))$
   - Reuse previous $\theta_k$ with probability: $\lambda_k(t_n) / (\lambda_0 + \sum_{k=1}^{K} \lambda_k(t_n))$

where $\lambda_k(t_n) = \sum_{i=1}^{n-1} \phi_{\theta_i}(t_n, t_i) \mathbb{I}(\theta_i = \theta_k)$ is the intensity function of a Hawkes process for past events with type $k$.

## 4   Hierarchical Dirichlet Point Process of Neural Sequences

We model the neural sequences as a hierarchical Dirichlet point process (HDPP). We use a Dirichlet nonhomogeneous Poisson process as the prior for observed spikes while using a Dirichlet Hawkes process as the prior for neural sequences. As in Figure 1, the goal of HDPP is to sequentially identify sequences from observed spikes, then partition these sequences into different types based on both temporal dynamics and included neurons.

Consider a continuous-time spike train data with $N$ neurons, its observed spikes can be denoted by a set of $S$ labeled events $x_s = (t_s, n_s, k_s)$ referring to the spike time, $t_s \in [0, T]$, neuron, $n_s \in \{1, ..., N\}$ and indicator to the sequence that $x_s$ belongs to, $k_s \in \{1, ..., K\}$. Note that the indicator $k_s$ remains unknown until we infer the sequences from the data. Similarly, the inferred neural sequences can also be represented as a collection of tuples $q_k = (\tau_k, m_k)$ referring to the time $\tau_k \in [0, T]$ and indicator for sequence type $m_k \in \{1, .., M\}$.

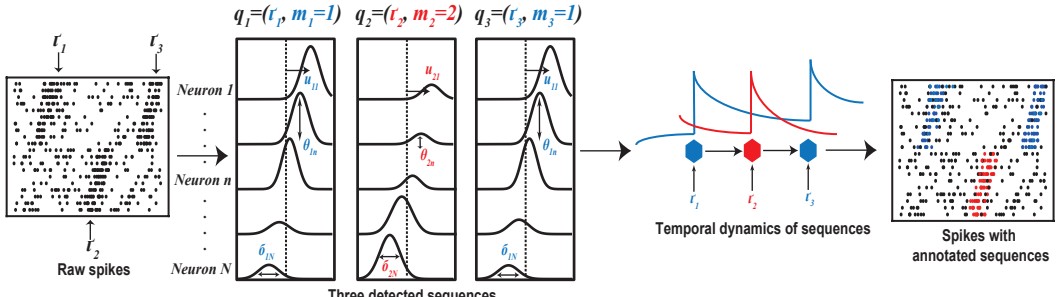

Figure 1: A case with $K = 3$ sequences and $M = 2$ types of sequences is shown. The proposed model considers observed spikes as a mixture of neural sequences, where the spikes of different neurons within a sequence can be modeled via a nonhomogeneous Poisson process that catches their firing rates. Likewise, the detected sequences are also a mixture of certain types of sequences, where their temporal dynamics are captured by several Hawkes processes.

According to the definition of the Dirichlet Hawkes process, for a spike $x_s$ occurring at time $t_s$ its probability of belonging to the $k$-th neural sequence is:

$$P(k_s = k | t_s) = \frac{g_k(t_s)}{g_0 + g_{\text{noise}} + \sum_{k=1}^{K} g_k(t_s)}, \tag{2}$$

where $g_0$ is the concentration parameter, $g_{\text{noise}}$ is the firing rate for background spikes, and $g_k(t)$ is the intensity function of sequence $q_k$, controlling the temporal dynamics of related spikes. Likewise, the probability of assigning spike $x_s$ to either background noise or a new sequence is given by:

$$P(k_s = k | t_s) = \begin{cases} \frac{g_{\text{noise}}}{g_0 + g_{\text{noise}} + \sum_{k=1}^{K} g_k(t_s)} & \text{for } k = 0, \\ \frac{g_0}{g_0 + g_{\text{noise}} + \sum_{k=1}^{K} g_k(t_s)} & \text{for } k = K + 1. \end{cases} \tag{3}$$

After a neural sequence $q_{k>0}$ is chosen, we assign it to an existing type $m_k$ or create a new type on the fly as needed with probability:

$$P(m_k = m | \tau_k) = \begin{cases} \frac{\lambda_m(\tau_k)}{\lambda_0 + \sum_{m=1}^{M} \lambda_m(\tau_k)} & \text{for } m \in \{1, ..., M\}, \\ \frac{\lambda_0}{\lambda_0 + \sum_{m=1}^{M} \lambda_m(\tau_k)} & m = M + 1. \end{cases} \tag{4}$$

where $\lambda_m(\tau)$ is the intensity function that captures temporal dynamics of sequences in type $m$. Note that we assume $g_k(t)$ and $\lambda_m(\tau)$ take different functional forms, and $g_{kn}(t)$ can be viewed as the firing rate of neuron $n$ caused by sequence $q_k$. Following the work in [28], we represent $g_{kn}(t)$ via a Gaussian form as:

$$g_{kn}(t) = \theta_{m_k n} \cdot \beta_k \cdot \mathcal{N}(t | \tau_k + \mu_{m_k n}, \sigma_{m_k n}). \tag{5}$$

Here $\theta_{m_k n}$, $\mu_{m_k n}$ and $\sigma_{m_k n}$ are parameters of each sequence type $m_k$, indicating the weight, offset, and width of neuron $n$'s firing rate. $\beta_k$ is the amplitude of sequence $q_k$, denoting the expected number of spikes induced by $q_k$ [28]. For $\lambda_m(\tau)$, we opt the similar form as (1), with parameter $\alpha_m > 0$ controlling the self-excitation of previous sequences, hyperparameter $\delta > 0$ controls the decay and hyperparameter $\eta$ denotes the expected time interval of sequences:

$$\lambda_m(\tau) = \alpha_m \sum_{\tau_k} \exp(-\delta(\tau - \tau_k - \eta)) \mathbb{I}(\theta_k = \theta_m). \tag{6}$$

Based on the definition of $g_k(t)$ and $\lambda_m(\tau)$, we regard that the temporal dynamics of spikes and neural sequences are captured by a nonhomogeneous Poisson process and a self-exciting Hawkes process, respectively. For sequence type parameters $\alpha_m$, $\{\theta_{mn}\}_{n=1}^{N}$ and $\{(\mu_{mn}, \sigma_{mn})\}_{n=1}^{N}$, we assume they follow a Gamma distribution, a Dirichlet distribution, and a normal-inverse-chi-squared distribution [17, 28] for every neuron and type. For amplitude $\beta_k$ and background spike rate $g_{\text{noise}}$, we assume they obey a Gamma distribution. Then, we can describe the proposed hierarchical Dirichlet point process in a generative way as:

1. For neural sequence $q_k$, $\tau_k \sim \text{HP}(\lambda_0 + \sum_{m=1}^{M} \lambda_m(\tau_{k-1}))$, $\beta_k \sim \text{Gamma}(a_1, b_1)$.
2. Sample sequence type $m_k$ using (4).
   - If $m_k = M + 1$: $\theta_{M+1} \sim \text{Dir}(\theta_0)$, $(\mu_{M+1}, \sigma_{M+1}) \sim \text{NI}\chi^2(\mu_0, \kappa_0, \upsilon_0, \sigma_0)$, $\alpha_{M+1} \sim \text{Gamma}(a_2, b_2)$ and $M = M + 1$.
3. Sample spikes induced by sequence $q_k$: $s_k = \text{Poisson}(\beta_k)$, $\{n_s\}_{s=1}^{s_k} \sim \text{Categorical}(\theta_{m_k})$ and $\{t_s\}_{s=1}^{s_k} \sim \mathcal{N}(\tau_k + \mu_{m_k n_s}, \sigma_{m_k n_s})$
4. Sample background spikes during time interval $\Delta t = \tau_k - \tau_{k-1}$: $g_{\text{noise}} \sim \text{Gamma}(a_0, b_0)$, $s_0 = \text{Poisson}(g_{\text{noise}} \cdot \Delta t)$, $\{n\}_{s=1}^{s_0} \sim \text{Categorical}(\theta_{\text{noise}})$ and $\{t_s\}_{s=1}^{s_0} \sim \text{Uniform}([\tau_{k-1}, \tau_k])$.

where $\theta_0$ is a hyperparameter parameter for Dirichlet prior, $a_0, b_0, a_1, b_1, a_2, b_2$ are hyperparameter parameters for Gamma prior, and $\mu_0, \kappa_0, \upsilon_0, \sigma_0$ are hyperparameter parameters for normal-inverse-chi-squared prior.

## 5 Inference

Given spikes $\{x_s = (t_s, n_s)\}_{s=1}^{S}$, as previously discussed, the model aims to sample neural sequences indicator $k_s$ for each spike. Using this insight, we can use a set of particles to sequentially approximate posterior $P(k_{1:s}|t_{1:s}, n_{1:s})$, while sampling indicator $m_k$ for each sequence.

We derive a Particle filter approach for our proposed model. This inference algorithm exploits the temporal dependencies in the observed spikes to sequentially sample the latent variables. To efficiently sample from the posterior distribution, a Particle filter keeps track of an approximation of $P(k_{1:s-1}|t_{1:s-1}, n_{1:s-1})$, and updates it to have an approximation for $P(k_{1:s}|t_{1:s}, n_{1:s})$ when receiving a newly observed spike. The approximation is evaluated by a set of weighted particles $\{p\}^P$, where the weight of each particle reflects how well the observed spikes are explained.

For each particle $p$, its weight is defined as the ratio between true posterior and importance distribution: $w_s^p = \frac{P(k_{1:s}|t_{1:s}, n_{1:s})}{Q(k_{1:s}|t_{1:s}, n_{1:s})}$. As the work in [1, 5] did, we choose $P(k_s|k_{1:s-1}, t_{1:s}, n_{1:s})$ to be the importance distribution. Next, the unnormalized weight $w_s^p$ can be updated as:

$$w_s^p \propto w_{s-1}^p \cdot P(n_s|k_{1:s}, n_{1:s-1}). \tag{7}$$

Here, let $C^{m_{k_s} \setminus x_s}$, $C_{n_s}^{m_{k_s} \setminus x_s}$ denote the spike counts of sequence type $m_{k_s}$ and spike counts of neuron $n_s$ in sequence type $m_{k_s}$, both excluding $x_s$. The likelihood $P(n_s|k_{1:s}, n_{1:s-1})$ is given by a Dirichlet-Categorical conjugate relation [16]:

$$\frac{C_{n_s}^{m_{k_s} \setminus x_s} + \theta_{0 n_s}}{C^{m_{k_s} \setminus x_s} + \sum_{n=1}^{N} \theta_{0n}}. \tag{8}$$

After sampling indicator $k_s$ from posterior $P(k_s|t_s, n_s, \text{rest})$ for each spike, we then sample the type indicators $m_k$ from posterior $P(m_k|\tau_k, X_k, \text{rest})$. The posteriors of both indicators are determined by the likelihood of the spike/sequence assigned to a given sequence/sequence type and by the temporal prior under Dirichlet nonhomogeneous Poisson process (2)(3) or Dirichlet Hawkes process (4):

$$P(k_s|t_s, n_s, \text{rest}) \propto P(n_s|k_s, \text{rest}) \cdot P(k_s|t_s, \text{rest}), \tag{9}$$

$$P(m_k|\tau_k, X_k, \text{rest}) \propto P(X_k|m_k, \text{rest}) \cdot P(m_k|\tau_k, \text{rest}), \tag{10}$$

where *rest* represents all other latent variables with $x_{1:s-1}$, $X_k = \{n_s : k_s = k\}$ is the neurons in $k$-th sequence. The exact form of the two posteriors is given in Supplement A.

**Updating parameters.** There are three groups of parameters that need to be updated after attributing each spike to a neural sequence: sequence type parameters $\Theta_m = (\theta_m, \alpha_m, \{\mu_{mn}\}_{n=1}^{N}, \{\sigma_{mn}\}_{n=1}^{N})$, sequence parameters $\Theta_k = \{\tau_k, \beta_k\}$ and background parameters $\Theta_\emptyset = g_{\text{noise}}$. Following the literature in Particle filter devoted to the estimation of an online parameter [2], for most parameters, we perform a single Gibbs sampling step to sample from their closed-form posterior under conjugate relation. Details about derivations are in Supplement B.

**Merge and Prune.** Generally speaking, neural sequences created during this sequential inferring procedure are often truly needed, as the decisions to add new sequences are based on the accumulated

knowledge from the past. However, there is still a chance that some sequences generated at early samples would become incorrect and that multiple sequences may be close in time. Thus, we introduce a mechanism to merge similar sequences and prune incorrect ones.

The similarity of two sequences $q_k$ and $q_{k'}$ can be defined in terms of the time difference between $\tau_k$ and $\tau_{k'}$, as $d(k, k') = |\tau_k - \tau_{k'}|$. We will merge sequences $q_k$, $q_{k'}$ and their associated sufficient statistics when $d(k, k') < \Delta_d$. We also check the variance of firing times for each neuron in every sequence. That is, we reserve the spikes which fire consecutively in the same neuron $n$: $\text{Var}(\{t_s\}_{s=1}^{s_n}) < \epsilon$, while removing those that appear to be sparsely distributed. Besides, we will remove an entire sequence if it has too few spikes after pruning. Note that there is no need to perform a such merge or prune at every iteration. Since the operation of merge sequences takes $O(K)$ and computing variance for each sequence takes $O(s_n \cdot N)$, we propose to examine similarity and variance at a fixed interval $O(i \cdot T)$ and only for the current sampled sequence.

The overall pseudocode for inference is presented in Algorithm 1, and its time complexity is $O(SP(K + M))$, where $S$ is the number of spikes, $P$ is the number of particles, $K$ is the number of sequences, and $M$ is the number of types. Our code is available at `https://github.com/WeihanLikk/Hierarchical-Dirichlet-Point-Process`.

---

**Algorithm 1:** Particle filter for HDPP

---

**Input:** Spikes $\{x_s = (t_s, n_s)\}_{s=1}^S$, hyperparameters and threshold $\epsilon$.
Initialize particle weight $w^p$ to $\frac{1}{P}$ for all $p \in \{1, .., P\}$;
**for** *each spike* $(t_s, n_s), s = 1, ..., S$ **do**
    **for** $p \in \{1, ..., P\}$ **do**
        Sample sequence indicator $k_s$ from (9);
        Sample sequence type indicator $m_k$ from (10);
        Update the parameters $\Theta_m$, $\Theta_k$, and $\Theta_\emptyset$;
        Update the particle weight by (7);
    **end**
    Normalize particle weight;
    **if** $\|w^p\|_2^{-2} < \epsilon$ **then**
        Resample particles
    **end**
**end**

---

## 6 Experiments

### 6.1 Synthetic Data

We created three kinds of synthetic data to investigate the robustness to noise, the effectiveness of temporal dynamics, and the ability to infer the number of types of sequences, respectively, of HDPP: (1) Two types of sequences with background noise. (2) Two types of sequences occur side-by-side with spatial overlapping neurons. (3) A large set of sequence types with spatial overlapping neurons, where they are much uncertainty in sequence types.

**Robustness to background noise.** Figure 2(a) presents the first case of synthetic data, which demonstrates the robustness of HDPP. This data has $N = 60$ neurons and $T = 30$ seconds. The raw spikes (Figure 2(a) left panel) show no visible neural sequences. However, sequences are revealed by sorting the neurons with the learned weight, type, and offset of this sequence. Besides, background spikes can also be correctly identified as noise.

**Distinguishing overlapping sequences.** Figure 2(b) shows a much more complex case where two sequences with overlap populations of neurons occur side-by-side. This data has $N = 60$ neurons and $T = 20$ seconds. Usually, it's hard to distinguish these sequences from the perspective of neurons. But with the help of temporal dynamics, HDPP could be able to capture sequences' distinctive intensity functions (Figure 2(b) top panel), and disambiguate them to different types. Note that the intensities at the very beginning are missing since HDPP resamples a type indicator for the first

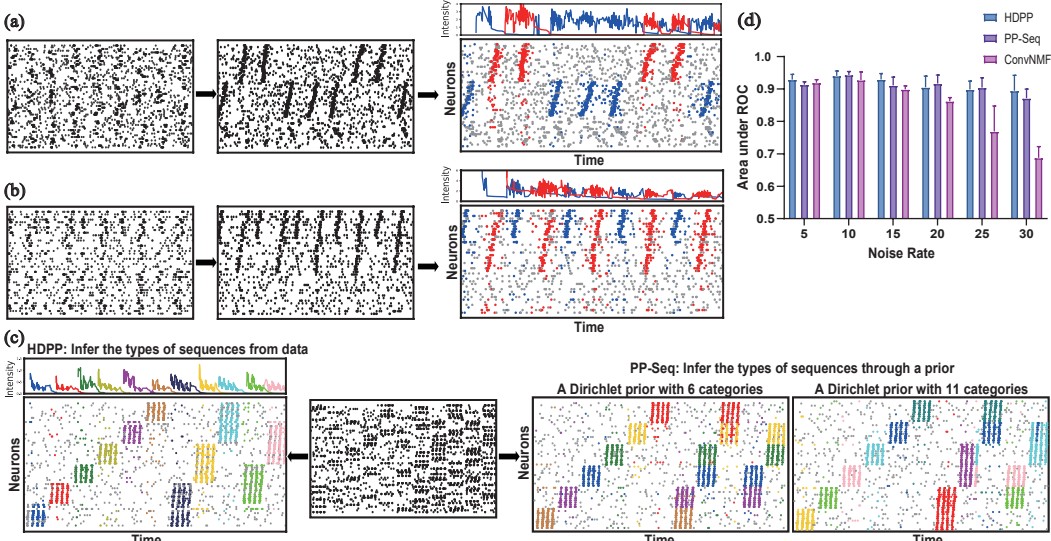

Figure 2: Synthetic data. (a) Two types of sequences with background noise. Raw spikes (left), neurons re-sorted by HDPP (middle), sequences annotated by HDPP (right), and corresponding intensities captured by Hawkes processes (top). (b) Two types of highly overlapping sequences occur side-by-side with background noise. (c) Comparison of HDPP and PP-Seq on a large set of sequence types with spatial overlapping neurons. HDPP can infer the types of sequences from data (d) Comparison of HDPP, PP-Seq, and convNMF to identify sequences under different levels of background noise. HDPP is robust to background noise.

sequence whenever it receives a spike and those empty types are dropped as well as their recorded intensities.

**Inferring the types of sequences from data.** Figure 2(c) shows the comparison of HDPP and PP-Seq, demonstrating that HDPP can infer the number of types from data. This data has $N = 60$ neurons and $T = 400$ seconds, where there are 11 different types of sequences: 6 types of smaller sequences involving 10 neurons, while other 5 types of larger sequences involving 20 neurons. Note that the larger sequences and smaller sequences have spatial overlapping neurons. HDPP could be able to distinguish the sequences with overlapping neurons without prior information on the number of types (Figure 2(c), left panel), while PP-Seq identifies the larger sequences as a new type when setting the prior of types to a Dirichlet prior with 11 categories (Figure 2(c), right panel).

Further, we also tested the robustness of HDPP quantitatively with the area under receiver operating characteristic (ROC) curves as a metric. To keep in line with previous work[28], we simulated a dataset with $M = 1$ sequence type and $N = 100$ neurons, where we varied the background spikes $g_{\text{noise}}$ to quantitatively measure the robustness. From the results of the area under ROC in Figure 2(d), HDPP achieves promising results as we increase the noise levels.

## 6.2 Performance with Neural Recordings

### 6.2.1 Songbird Higher Vocal Center Recording

For real-world data, we first applied our model to the deconvolved spike train from functional imaging data recorded in songbird HVC during singing [15][2]. One of the core features of this data is the high variability of the song, thus making it difficult to identify neural sequences by a template-matching procedure.

**Identifying neural sequences from Songbird Recording.** Figure 3(a) visualizes the results from HDPP and PP-Seq[3], where we chose the neurons whose weight $\{\theta_{mn}\}_{n=1}^{N}$ was above a threshold, and sorted them based on each sequence's type $m_k$ and offset $\{\mu_{mn}\}_{n=1}^{N}$. Figure 3(b)-(c) visualize

---

[2] http://github.com/FeeLab/seqNMF
[3] https://github.com/lindermanlab/PPSeq.jl

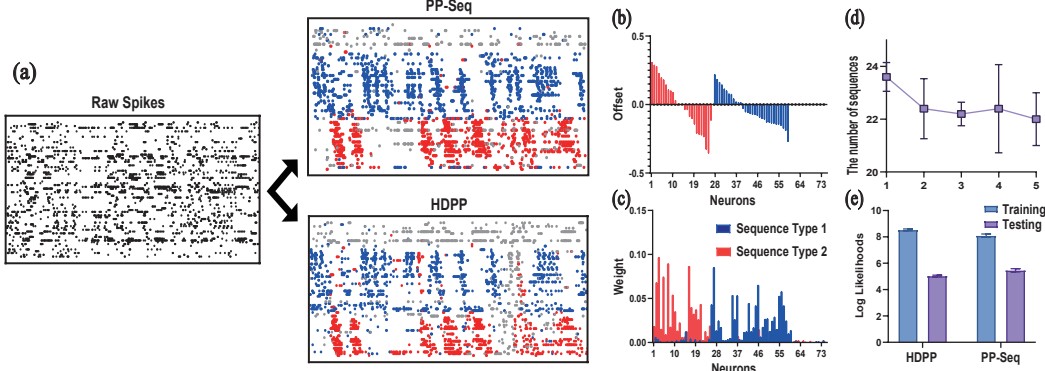

Figure 3: Songbird Higher Vocal Center data. (a) Raw spike train (left) and comparison of detected sequences by HDPP (bottom) and PP-Seq (top). (b)(c) Visualization of learned offsets and weights for sequence type 1 (blue) and sequence type 2 (red). (d) The number of detected sequences over five independent runs. (e) The log-likelihoods comparison of HDPP and PP-Seq.

the learned offset and weight from HDPP. For offset $\{\mu_{mn}\}_{n=1}^N$, there is a clear time-shift among different neurons ($\pm$ 0.25s for type 1 and $\pm$ 0.3s for type 2), and for weight $\{\theta_{mn}\}_{n=1}^N$, the two sequence types have their own preference for neurons (neuron 27~62 for type 1 and neuron 1~26 for type 2). Figure 3(d) indicates HDPP converges to similar parameter ranges: The total number of detected sequences over five independent runs (different random seeds for initiation) are close to each other. Finally, Figure 3(e) shows the log-likelihood comparison of HDPP and PP-Seq averaging over different runs.

### 6.2.2   Rodent Hippocampal Recording

We then tested our model on a more complex spike train data from a rat's hippocampal recording when it performs repeating runs down a wooden 1.6m linear track [3, 8, 7][4]. The session we used is "Achilles 10252013", which has $N = 120$ neurons and $T = 2068$ seconds. As in [8], we expect to observe two sequences encoding the two opposite running directions on this liner track. To have a good knowledge of hyperparameter settings, we performed cross-validation via a "speckled holdout pattern" strategy, which was previously used in PCA [33], and has recently been applied to neural data [28, 29]. We then evaluated the performance of HDPP by calculating log-likelihood on the held-out set.

**Identifying neural sequences from Hippocampal Recording.** Figure 4(a)-(c) shows the raw spikes, sorted results, and two types of sequences annotated by HDPP, where we can clearly observe two groups of place coding neurons revealed by the sequences. Figure 4(d) presents the validation log-likelihoods of four key hyperparameters in HDPP, while other parameters have less impact on performance or are easy to locate in the optimal range. For the sequence amplitude $\beta_k$, we randomized the mean sequence amplitude $\frac{a_1}{b_1}$ and set the variance of sequence amplitude $\frac{a_1}{b_1^2}$ equals to the mean. For the background noise amplitude, we randomized the prior rate $b_0$ in gamma distribution and set the prior shape $a_0$ to 100. According to the results, HDPP on this data prefers larger sequence amplitude $\beta_k$ (higher mean sequence amplitude $\frac{a_1}{b_1}$), higher width $\sigma_0$ in the NI$\chi^2$ distribution, lower noise firing rate $g_{\text{noise}}$ (lower mean noise firing rate $\frac{a_0}{b_0}$), and higher concentration parameter $g_0$ in the Dirichlet nonhomogeneous Poisson process.

### 6.3   Scalability to Large Dataset

To scale with large recordings, HDPP is supposed to process each spike in an expected constant time. In other words, the time cost of either calculating intensity functions or updating parameters should not grow with the number of sequences detected. In HDPP, we can note that the influences of past sequences decay exponentially. Using this insight, we can reduce the computational load by ignoring those past sequences far away from the current time. More specifically, given a time window $\Delta t$, we

---

[4]`http://crcns.org/data-sets/hc/hc-11`

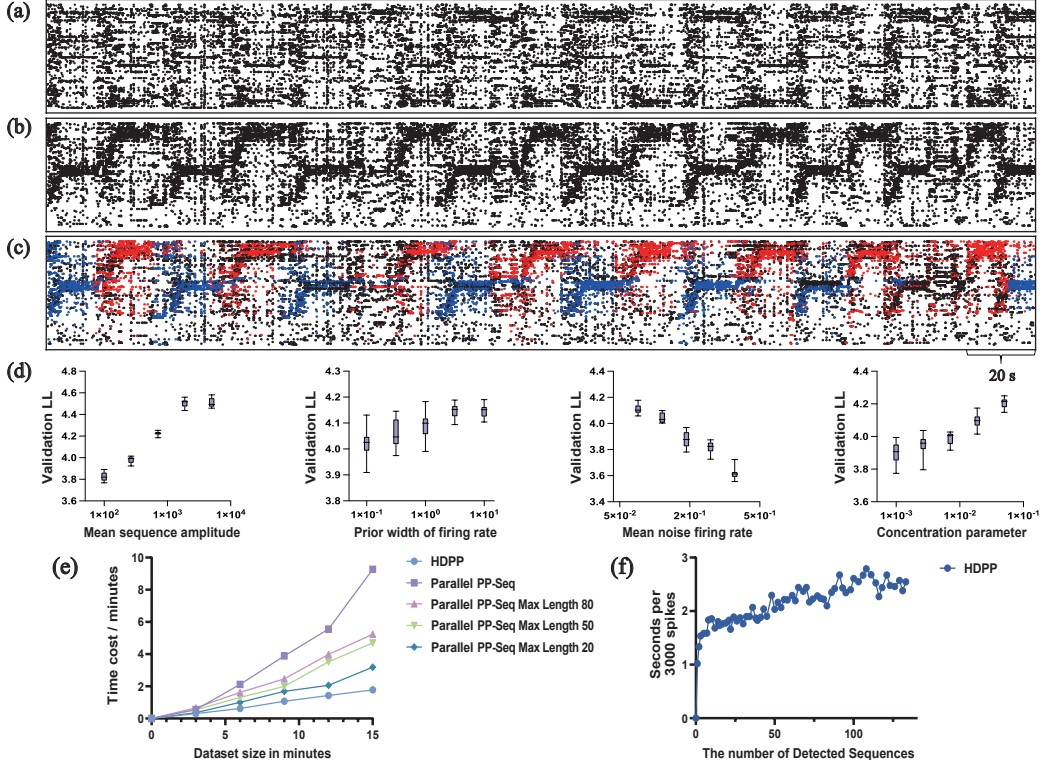

Figure 4: Rodent hippocampal recording. (a) Raw spike train ($T \approx 4.6$ minutes is shown). (b) Neurons sorted by HDPP. (c) Two types of sequences identified by HDPP. (d) The validation log-likelihoods on four key hyperparameters. Each boxplot involves five different values of a hyperparameter, while other hyperparameters are set to fixed values. (e) Performance comparison between HDPP and parallel PP-Seq on rodent hippocampal recording. (f) HDPP's time cost as a function of detected sequences.

maintain an active list of sequences $\{q_k\}_{k=1}^K$ where $t - \tau_k \leq \Delta t$. Similarly, using temporal windows is also acceptable for PP-Seq by limiting the max length of each sequence, which is supposed to reduce its loop operations. The test of performance was carried by a desktop with AMD Ryzen 7 5800X 8-Core Processor and 32 GB RAM. Figure 4(e) shows the performance comparison between HDPP and PP-Seq on rodent hippocampal recording. For HDPP, we set the particle number to 20, and for PP-Seq, we adopt parallel MCMC to parallelize the computation as well as setting the number of Gibbs sweeps to 2100 [28]. The results indicate that HDPP is still faster than parallel PP-Seq though it benefits from the temporal window. Figure 4(f) presents the time cost as a function of detected sequences, which empirically verifies that HDPP has a constant time cost per spike after settling in.

## 7    Discussion

We present a hierarchical Dirichlet point process (HDPP) for neural sequence detection with an efficient online inference algorithm. The proposed model follows the framework of the Dirichlet process, where the spike firing rates of a sequence are modeled as a nonhomogeneous Poisson process, and the temporal dynamic of this sequence is captured via a Hawkes process. Unlike previous works, HDPP learns the number of sequence types adaptively during inference and takes in data in a streaming fashion, thus can be easily adapted to new data or large neural recordings. In the experiments, favorable results from both synthetic data and real-world spike recordings have demonstrated that HDPP can reliably and efficiently identify neural sequences in a single pass.

However, our model may fail in some conditions: (1) Time warping noise. Our model does not model the variability in sequence, which is a common phenomenon in neural data [28, 31]. (2) A Low firing rate of neurons. Generally speaking, a high firing rate makes identifying sequences from background

noise easier. As the firing rate becomes lower, our model's and existing methods' performance decreases (supplement material Figure 1). Furthermore, it is an exciting future direction to investigate how we can learn HDPP via online variational inference without truncated approximation of the stick-breaking construction [13, 27].

## 8 Acknowledgements

This work was partly supported by grants from the China Brain Project (2021ZD0200400), and Natural Science Foundation of China (U1909202, 61906166, 61925603), and the Key Research and Development Program of Zhejiang Province in China (2020C03004).

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
