## A  Posterior

The posterior of indicator $k_s$ is:

$$P(k_s|t_{1:s}, n_{1:s}, k_{1:s-1}) \propto P(n_s|k_{1:s}, n_{1:s-1}) \cdot P(k_s|t_{1:s}, k_{1:s-1})$$

$$\propto \frac{C_{n_s}^{m_{k_s}\backslash x_s} + \theta_{0n_s}}{C^{m_{k_s}\backslash x_s} + \sum_{n=1}^{N} \theta_{0n}} \cdot P(k_s|t_s, k_{1:s-1}), \tag{1}$$

where $P(k_s|t_{1:s}, k_{1:s-1})$ is the prior given by Dirichlet nonhomogeneous Poisson process:

$$P(k_s = k|t_{1:s}, k_{1:s-1}) = \begin{cases} \frac{g_{\text{noise}}}{g_0 + g_{\text{noise}} + \sum_{k=1}^{K} g_k(t_s)} & \text{for } k = 0, \\ \frac{g_k(t_s)}{g_0 + g_{\text{noise}} + \sum_{k=1}^{K} g_k(t_s)} & \text{for } k \in \{1, ..., K\}, \\ \frac{g_0}{g_0 + g_{\text{noise}} + \sum_{k=1}^{K} g_k(t_s)} & \text{for } k = K+1. \end{cases} \tag{2}$$

The posterior of indicator $m_k$ is:

$$P(m_k|\tau_{1:k}, X_{1:k}, m_{1:k-1}) \propto P(X_k|m_{1:k}, X_{1:k-1}) \cdot P(m_k|\tau_{1:k}, m_{1:k-1})$$

$$\propto \prod_{x_s \in X_k} \frac{C_{n_s}^{m_k\backslash x_s} + \theta_{0n_s}}{C^{m_k\backslash x_s} + \sum_{n=1}^{N} \theta_{0n}} \cdot P(m_k|\tau_k, m_{1:k-1}), \tag{3}$$

where $P(m_k|\tau_{1:k}, m_{1:k-1})$ is the prior given by Dirichlet Hawkes process:

$$P(m_k = m|\tau_{1:k}, m_{1:k-1}) = \begin{cases} \frac{\lambda_m(\tau_k)}{\lambda_0 + \sum_{m=1}^{M} \lambda_m(\tau_k)} & \text{for } m \in \{1, ..., M\}, \\ \frac{\lambda_0}{\lambda_0 + \sum_{m=1}^{M} \lambda_m(\tau_k)} & m = M+1. \end{cases} \tag{4}$$

## B  Parameter Updates

### B.1  Sequence type parameters $\Theta_m$

For weight $\theta_m$, its posterior is given by the Dirichlet-Categorical relation:

$$P(\theta_m|X_{1:K}) \propto \text{Dir}(\theta_m|\theta_0) \cdot \prod_{k=1}^{K} \prod_{x_s \in X_k} \text{Categorical}(n_s|\theta_{m_k})$$

$$\propto \text{Dir}(\theta_m'), \tag{5}$$

where $\theta_{mn}' = \theta_{0n} + \sum_{k=1}^{K} \sum_{x_s \in X_k} \mathbb{I}(n_s = n, m_k = m)$.

For kernel parameter $\alpha_m$, its posterior is given by the Gamma-Exponential relation:

$$P(\alpha_m|X_{1:K}, \{\Theta_k\}_{k=1}^{K}) \propto \text{Gamma}(a_2, b_2) \cdot \alpha_m^{k'} \exp(-\frac{\alpha_m}{\delta} \sum_{k=1}^{k'} (\exp(\delta\eta) - \exp(-\delta(\tau_{k'} - \tau_k - \eta))))$$

$$\propto \text{Gamma}(a_2 + k', b_2 + \frac{1}{\delta} \sum_{k=1}^{k'} (\exp(\delta\eta) - \exp(-\delta(\tau_{k'} - \tau_k - \eta)))), \tag{6}$$

where $k' = \sum_{k=1}^{K} \mathbb{I}(m_k = m)$.

For offset and bias parameter $\mu_{mn}, \sigma_{mn}$, their posterior are given by the normal inverse chi squared-Univariate Normal relation [1, 2]:

$$P(\mu_{mn}, \sigma_{mn}|X_{1:K}, \{\Theta_k\}_{k=1}^{K}) \propto \mathcal{N}(\mu_{mn}|\mu_0, \frac{\sigma_{mn}}{\kappa_0}) \cdot \chi^{-2}(\sigma_{mn}|\upsilon_0, \sigma_0) \cdot$$

$$\prod_{k=1}^{K} \prod_{x_s \in X_k} (\mathcal{N}(t_s|\tau_k + \mu_{mn}, \sigma_{mn}))^{\mathbb{I}(n_s=n, m_k=m)} \tag{7}$$

$$\propto \text{NI}\chi^2(\mu_{mn}, \sigma_{mn}|\mu_{mn}', \kappa_{mn}, \upsilon_{mn}, \sigma_{mn}')$$

where let $Z = \{t_s - \tau_k : x_s \in X_k, n_s = n, m_k = m\}$, $\overline{Z}$ be the mean of $Z$, and we have

$$S_{mn} = \sum_{k=1}^{K} \sum_{x_s \in X_k} \mathbb{I}(n_s = n, m_k = m)$$

$$\kappa_{mn} = \kappa_0 + S_{mn}$$

$$v_{mn} = v_0 + S_{mn} \tag{8}$$

$$\mu'_{mn} = \frac{S_{mn}\overline{Z}}{\kappa_{mn}}$$

$$v_{mn}\sigma'_{mn} = v_0\sigma_0 + \sum_{s=1}^{S_{mn}} (Z_s - \overline{Z})^2 + \frac{S_{mn}\kappa_0}{\kappa_0 + S_{mn}}(\overline{Z})^2$$

### B.2    Sequence parameters $\Theta_k$

For sequence amplitude $\beta_k$, its posterior is given by the Gamma-Poisson relation:

$$P(\beta_k|X_k) \propto \text{Gamma}(a_1, b_1) \cdot \text{Poisson}(s_k|\beta_k)$$
$$\propto \text{Gamma}(a_1 + s_k, b_1 + 1) \tag{9}$$

where $s_k$ is the current number of spikes in sequence $k$.

For sequence time $\tau_k$, there is no conjugate relation between the prior of $\tau_k$ and likelihood of $\{t_s : x_s \in X_k\}$, and Metropolis-within-Gibbs sampling steps are usually required to sample from its posterior. However, this sampling is inefficient and we choose to update its value by taking the average of related spike times $\{t_s : x_s \in X_k\}$, which is a rough approximation but is useful in practice.

### B.3    Background parameters $\Theta_\emptyset$

For background noise rate $g_{\text{noise}}$, its posterior is given by the Gamma-Exponential relation:

$$P(g_{\text{noise}}|X_0) \propto \text{Gamma}(a_0, b_0) \cdot g_{\text{noise}}^{s_0} \exp(-g_{\text{noise}} \cdot \Delta t)$$
$$\propto \text{Gamma}(a_0 + s_0, b_0 + \Delta t) \tag{10}$$

where $s_0$ is the current number of background spikes and $\Delta t$ is the time duration from the very beginning.

## C    Additional Figures for Experiment

### C.1    Synthetic data with Low Firing Rate

In addition to the background noise, we also test the performance of HDPP, PP-Seq and ConVNMF when the firing rates of neurons are low. In general, a low firing rate makes it harder to identify neural sequences from background noise. According to the results in Figure 1, as the firing rate becomes lower, the performance of our model and existing methods decreased.

For the computation of ROC curves, we follow the steps in the Section F.1 of PP-Seq's supplementary material [2] to compute ROC curves.

### C.2    Synthetic data with Spatial Overlapping Neurons

In addition to the overlapping experiment in Sec 6.1 of the main text, we also test the performance of our method under higher spatial overlapping neurons (75% overlapping). From the result in Figure 2, we note that our method still detects the two different types of sequences.

### C.3    Memory Comparison on Rodent Hippocampal Recording

Compared with parallel PP-Seq, our method has a much fewer time cost, but higher memory allocations, which is believed to be one of the causes to increase the time cost of our method (Figure 3). We think the cause of such high memory allocation comes from particle resampling, which involves many copy operations to transfer sufficient statistics among particles.

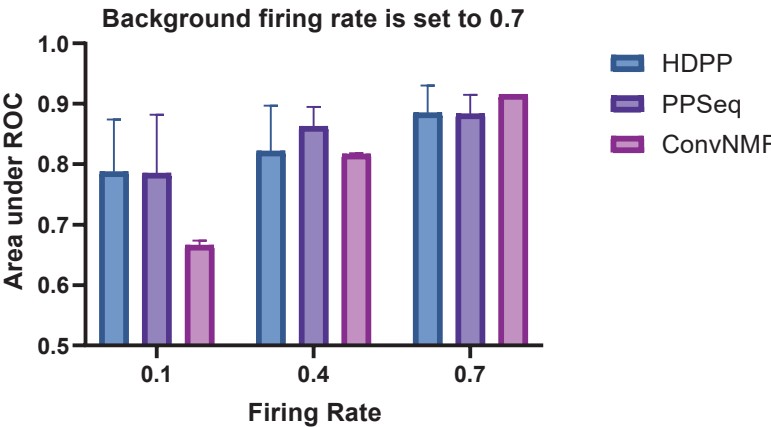

Figure 1: Performance of HDPP, PP-Seq and ConVNMF when the firing rates of sequences are low

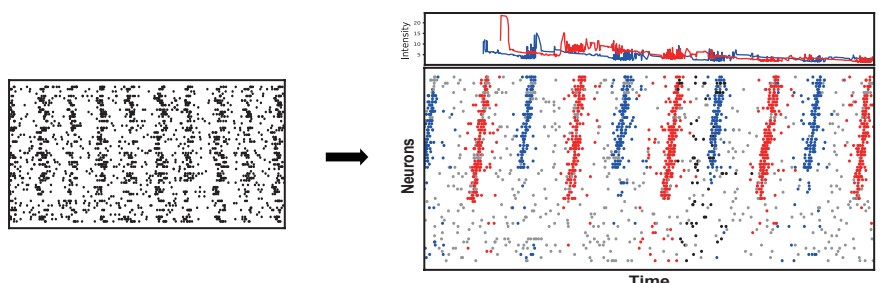

Figure 2: Synthetic data with 75% spatial overlapping neurons

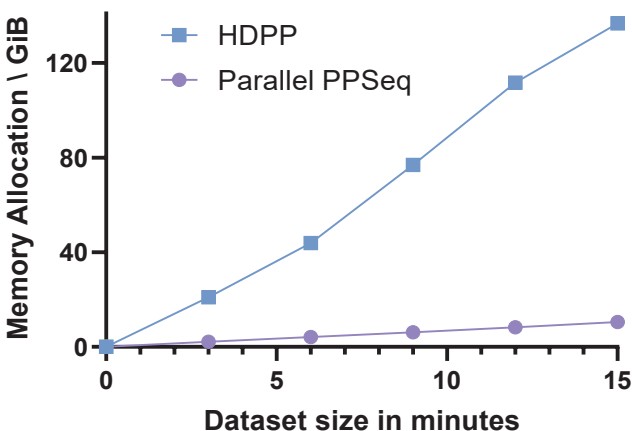

Figure 3: The memory cost of HDPP and PP-Seq on rodent hippocampal recording.

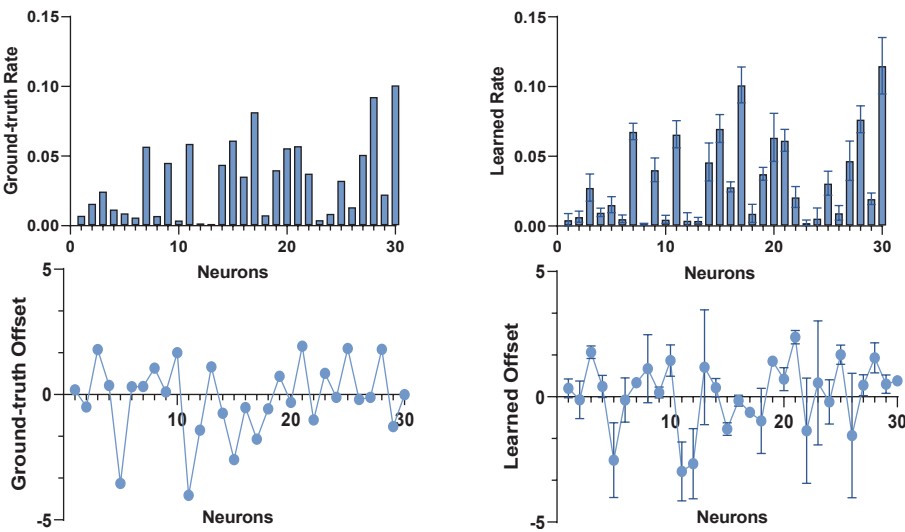

Figure 4: Analysing the convergence by recovering ground-truth parameters.

## C.4 Analysing the convergence by recovering ground-truth parameters

We create a synthetic data ($N = 30$, $K = 5$, $M = 1$) via the generating process described in section 4. The left two panels in Figure 4 show the visualization of ground truth parameters and the right two panels show the rate and offset for each neuron with $95\%$ credible intervals (calculate over five runs), which further demonstrates that our method could stably recover ground-truth parameters over different runs.

## D Broader Impact

Understanding human brain has wide implications and is a longstanding challenge in neuroscience research. Neural sequences have been observed in many neuroscience experiments and the detection of them plays an important role in discovering neural circuits and studying neural computation. Therefore, establishing useful analytical tools for neuroscientists is of vital importance for a better understanding of the human brain. We expect our model could serve to this challenge by providing an online, unsupervised approach to identify neural sequences.