# OpenReview forum: "Online Neural Sequence Detection with Hierarchical Dirichlet Point Process"
_NeurIPS.cc/2022/Conference — NeurIPS 2022 Accept_

### Official Review · Reviewer_L7EJ · 2022-07-02

**Rating:** 6
**Confidence:** 4
**Soundness:** 3 good
**Presentation:** 2 fair
**Contribution:** 2 fair

**Summary:**

One hypothesis about neural spike data is that it can be considered as a sequence of 'neural syllables' consisting of the spiking activity of a subset of neurons over a short period of time. Understanding these sequences would then enable relating activity recordings to behaviorally or experimentally relevant observables. The present manuscript proposes a method for unsupervised identification of such sequences.

Recent ML literature, as cited in the manuscript, studied similar problems for point processes in different contexts. In that sense, this manuscript primarily represents an application of existing methodology on hierarchical inference in temporally structured data. Briefly, the manuscript proposes a particle filter based (sampling-based) solution to inferring sequence labels in a hierarchical model where the observables are samples from point processes. The problem is both important and relevant to this community.

The manuscript uses a Dirichlet process prior, which enables the method to decide on the number of sequence labels without relying on prior information. Indeed, such information is not easy to know in practice. Briefly, the sequence types (labels) are modeled as a Dirichlet process so that the same type can appear multiple types in a given recording. Neuronal spiking activity is then modeled as a Hawkes process conditioned on the sequence type. Overall, the model is complicated, but it is fitting to the underlying complicated problem.

Update on Aug. 8: In response to the author rebuttal, I decided to increase my overall score from 5 to 6.

**Questions:**

I listed a few questions under Weaknesses above. I'd like to add a few more questions here:

- Could Fig. 4 be turned into a comparative study as well? How much time would previous methods require to process that dataset? I'd be curious to see the results of other methods after temporal windowing as I mentioned under Weaknesses.

- Could you please sort the neurons in Fig. 3 identically across the two compared methods to facilitate comparison?

- Could you please discuss how the different methods would perform when the firing rates are lower?

- Overall, the demonstrations use only a few sequence types. This may be in line with previous literature. However, one might expect to see a much larger set of sequence types in larger, longer recordings. Could you please discuss how the different methods would perform when the number of sequence types is larger?

**Limitations:**

The authors should discuss technical limitations of the work. That is, under which conditions can we expect it to fail? What are the roles of different noise sources, firing rates, etc?

**Strengths And Weaknesses:**

Strengths:
- Unsupervised, adaptive (streaming) inference of the number of sequence types
- Computational scalability

Weaknesses:
- Poor presentation: The manuscript is full of typos and grammatical mistakes. While this obscures the underlying meaning very rarely, reading the manuscript becomes time-consuming and frustrating. Similarly, reasoning should be better and more precisely explained. For instance, the argument in lines 266-267 on using temporal windows to reduce computational load appears applicable to all methods. Please better explain why it would be more beneficial for the proposed method compared to existing methods.

- Other than unsupervised inference of the number of types, one potential advantage of the proposed method could be real-time detection of sequences in a closed-loop experiment if inference can be made faster. (According to Fig. 5, it doesn't appear fast enough with the hardware used in that experiment.) However, this point was never discussed. Is this a desirable capability for neuroscience? What kind of desirable experiments could be performed by such a closed-loop setup? In my opinion, this point determines whether the proposed contribution is timely and impactful or a technical upgrade without practical consequences.

---

> ### Author Response · Authors · 2022-08-02
> **Response**
>
> Thank you for the appreciation and constructive comments. We have made several improvements according to your questions and comments. We hope these sufficiently clarified your concerns, and that they can be taken into account when deciding the final review score.
>
> The main modifications are as follows:
>
> * Extra experiments were carried out:
>    * We applied temporal windows to PP-Seq, and compared its time cost with our method on the hippocampal data in Figure 4(e).
>    * We explored the performance of our method, PP-Seq, and ConvNMF when the firing rates are lower in supplement material Figure 1.
>    * We explored the performance of our method and PP-Seq when the number of sequence types is large in Figure 2(c).
> * We have added the limitation of our approach in the Discussion.
>
> With these modifications, the soundness and presentation of the revised paper has improved.
>
> > The argument in lines 266-267 on using temporal windows to reduce computational load appears applicable to all methods. I'd be curious to see the results of other methods after temporal windowing.
>
> * We thank the reviewer for this suggestion, and extra experiments were carried out to apply a temporal window to limit the max length of sequence for the baseline method PP-Seq (Figure 4(e)). For a fair comparison, we reimplemented our method via Julia programming language, by which PP-Seq was implemented. From the results, our method is still faster than parallel PP-Seq though it is benefited from the temporal window.  Here, we report part of computation times (in seconds) of our method and PP-Seq:
>
> |  Hippocampal Recording length     | 3-minutes | 6-minutes | 9-minutes | 12-minutes | 15-minutes |
> | :---        |    :----:   |          ---: |   :----:   |   :----:   |   :----:   |
> | HDPP      | 18.2 | 37.3  | 64.5 | 85.5 | 106.6 |
> | Parallel PP-Seq    | 33.1   | 127.4 | 233.4 | 333.1 | 556.3 |
> | Parallel PP-Seq with max seq length 20   | 21.9 | 60.2 | 101.4 | 123.7 | 192.3 |
>
> >  Could Fig. 4 be turned into a comparative study as well? How much time would previous methods require to process that dataset?
>
> * As suggested, extra experiments were carried out to evaluate the time costs comparison of our method vs parallel PP-Seq on the hippocampal data in Figure 4(e) of the revised paper. According to the results, our method has a lower time cost compared with parallel PP-Seq.
>
> > Could you please sort the neurons in Fig. 3 identically across the two compared methods to facilitate comparison?
>
> * We thank the reviewer for this comment, and we have modified Figure 3 in the revised paper.
>
> > Could you please discuss how the different methods would perform when the firing rates are lower?
>
> * Extra experiments were carried out to evaluate the performance at lower firing rate of neurons. Please see Figure 1 in supplementary material. According to the result, as the firing rate becomes lower, the performance of all three approaches decreased. Overall, our method performed similar to PP-Seq, while was superior compared with ConvNMF.
>
> > Could you please discuss how the different methods would perform when the number of sequence types is larger?
>
> * Extra experiments with a new synthetic data were carried out as in the revised paper. Please refer to Figure 2 and Sec 6.1. This synthetic dataset has 11 different types of sequences with overlapping neurons: 6 types of smaller sequences contain 10 neurons, while the other 5 types of larger sequences contain 20 neurons. We compared our method with PP-Seq. According to the results, our method can identify all types of sequences without prior information on the number of types, while PP-Seq identifies the larger sequences when we set the number of types of sequences to 11.
>
> > One potential advantage of the proposed method could be real-time detection of sequences in a closed-loop experiment if inference can be made faster. What kind of desirable experiments could be performed by such a closed-loop setup?
>
> * We reimplemented our method via Julia programing language, and made it much faster than before. Our approach enables the online detection of neural sequences, which can be useful in diverse situations. For example, it can guide the electrode implantation process to check whether the target types of neurons exist or not around the implemented microelectrode array via identifying the sequences in a streaming way.
>
> > The authors should discuss technical limitations of the work.
>
> * We thank the reviewer for this suggestion and we have updated the paper to address the technical limitations of our method in Discussion.
>
> We again thank the reviewer for the helpful comments and constructive feedbacks which greatly help improve the quality of our paper. Many thanks.

---

> > ### Comment · Reviewer_L7EJ · 2022-08-08
> > **clarifying question**
> >
> > Thanks for the additional experiments and explanation. I think these have improved the paper. I have a related clarifying question: in the new Fig. 2c, PP-Seq appears to be given the number of sequence types explicitly (e.g., 6, 11). However, the original paper seems to infer this number through a prior, and not rely on the exact value of this parameter. Could you please explain?

---

> > > ### Author Response · Authors · 2022-08-08
> > > **Response**
> > >
> > > Many thanks for helping improve the paper and for revising the score! Indeed, PP-Seq does not require the number of sequence types explicitly but infers this number through a Dirichlet Prior with $R$ categories. Thanks for pointing it out. We have revised our paper to clarify this point.

---

### Official Review · Reviewer_Pow4 · 2022-07-08

**Rating:** 3
**Confidence:** 3
**Soundness:** 2 fair
**Presentation:** 2 fair
**Contribution:** 2 fair

**Summary:**

This paper proposed a neural sequence detection method, which is based on a so called ''hierarchical Dirichlet point process''. The model can jointly infer an unbounded number of sequences and the event types from data. The paper develops an online algorithm to process the neural spike data, with the particle filtering framework. The proposed method was evaluated on two real-world datasets.

**Questions:**

see above

**Ethics Review Area:**

["I don’t know"]

**Strengths And Weaknesses:**

Strengths:
1. The application is very interesting and important
2. The experimental data is interesting. From the test log-likelihood, the proposed algorithm seems working.

Weakness: In summary, there is a big issue in the presentation and the contributions seem weak.
1. The problem formulation is not clear. A good paper should give a clear definition and motivation of the problem you want to solve, but i did not see it in this paper, making me very confused.  Why do you want to partition the neural spikes into sequences? What is the meaning of doing so?  Aren't the observed events already a sequence by time? For a new sequence (inferred by the model), based on what the events are ordered? The notations confuse me further --- according to line 118, on the spike-train data, you've already had k_s to indicate the neural sequence, why do you want to infer them again with the model (see Eq2)?

2. The relationship with the classical HDP is not mentioned. The name is confusing in that I thought you are building a model similar to  the HDP of (Teh et. al. 2005). However, it is not. The classical HDP uses the first-level DP to sample an infinite set of bases, and each DP in the second level will share these bases, but generate the data with different mixture weights. It took me a while to find out that this paper is not following the HDP framework. Rather, it is more like a hybrid of two DPs, one is to partition the data into sequences, and the other to partition the events into types. I do not see a clear ``hierarchical'' structure here. The paper should highlight the difference with the classical HDP, to avoid unnecessary misunderstanding.

2. The contributions seem weak. The proposed model is  an extension of Dirichlet Hawkes process by (Du et. al., 2015). One more DP is added, and it is quite incremental. The inference, based on the particle filtering, is almost the same as the SMC used by (Du et. al., 2015), plus some heuristics to merge and prune. If there are some more significant contributions, the authors should highlight them. Otherwise, I view this is a very incremental extension of   (Du et. al., 2015), applied to spike-train data analysis.

3. The quantitative metric is only based on test log-likelihood. I have a lot of experience of doing point process modeling. Although many works use test log-likelihood for evaluation, it is not a reliable metric, and is quite misleading. I look forward to seeing some evaluation that can truly reflect the performance, such as future event time prediction.

---

> ### Author Response · Authors · 2022-08-02
> **Response**
>
> Many thanks for the feedback and constructive comments. We have given a point-to-point response to your comments below. Hopefully these will address most of your concerns, and can be taken into consideration when deciding the final review score of the paper.
>
> > Why do you want to partition the neural spikes into sequences? What is the meaning of doing so?
>
> * Detection of repetitive neural sequences is an important problem in neuroscience research because they help neuroscientists understand the brain activities such as learning[1], motor production[2], and working memory [3]. Our work will serve to advance this growing understanding by providing new analytical tools for theorists. We have revised the introduction of the paper to clearly show the definition and motivation of this topic.
>
> > Aren't the observed events already a sequence by time?
>
> * Indeed, spikes are already a sequence by time, but the neurons are not well ordered so we cannot directly observe neural sequences from data, thus we reorder the neurons to intuitively reflect the patterns, as in [4-5]. Specifically, since each spike event is represented by a tuple (neuron, time), we can sort the neurons to order the events. Firstly, the neurons that have low firing rates (few spike events) are excluded. Secondly, the neurons that have the same type of sequences are grouped. Finally, the neurons within each group are sorted by the inferred offset parameter $\mu_{mn}$ of this sequence type. We have added extra descriptions to make it clear in the revised paper.
>
> > According to line 118, on the spike-train data, you've already had k_s to indicate the neural sequence, why do you want to infer them again with the model (see Eq2)?
>
> * Since we cannot directly observe sequences from spikes, the sequence indicator $k_s$ of each spike remains unknown until we infer the sequence from the data. We have clarified it in the revised paper.
>
> > The relationship with the classical HDP is not mentioned. The paper should highlight the difference with the classical HDP, to avoid unnecessary misunderstanding.
>
> * We are grateful that the reviewer pointed out that “hierarchical” leads to unnecessary misunderstanding. Like the classical HDP, our method also has a two-level structure: the DP at the lower level identifies the sequences from spikes, and the DP at the higher level partitions the sequences into types. The only difference is we use only one DP in the lower level since we focus on a single-trial analysis in this paper. We have clarified the difference with the classical HDP in the revised paper.
>
> > The contributions seem weak.
>
> * The main contribution lies in that we propose an online spike train pattern detection approach, which to the best of our knowledge, is the first method that enables sequential identification of sequences from observed spikes and copes with new sequence types on the fly. Our method performs comparably to the state-of-the-art with a significantly lower time cost. The two-level DPs to neural sequences detection method is specifically proposed for this objective. The lower level Dirichlet Poisson process with a Gaussian form intensity function could correctly model the within the structure of sequences, and the higher level Dirichlet Hawkes process could greatly help disambiguate spatial overlapping sequences.
>
> > The quantitative metric is only based on test log-likelihood. I look forward to seeing some evaluation that can truly reflect the performance, such as future event time prediction.
>
> * We used the log-likelihood metric following the existing neural sequences detection approach [5]. We agree the future event time prediction ability is also an important metric, but it is difficult to carry out the experiments given time limitation. We are working on it and will add the suggested metric in the future work.
>
>
>
>
> We again thank the reviewer for the helpful comments and constructive feedbacks which greatly help improve the quality of our paper. Many thanks.
>
> **References**
>
> [1] Eichenbaum, Howard. "Time cells in the hippocampus: a new dimension for mapping memories." Nature Reviews Neuroscience 15.11 (2014): 732-744.
>
> [2] Hahnloser, Richard HR, Alexay A. Kozhevnikov, and Michale S. Fee. "An ultra-sparse code underliesthe generation of neural sequences in a songbird." Nature 419.6902 (2002): 65-70.
>
> [3] Goldman, Mark S. "Memory without feedback in a neural network." Neuron 61.4 (2009): 621-634.
>
> [4] Mackevicius, Emily L., et al. "Unsupervised discovery of temporal sequences in high-dimensional datasets, with applications to neuroscience." Elife 8 (2019): e38471.
>
> [5] Williams, Alex, et al. "Point process models for sequence detection in high-dimensional neural spike trains." Advances in neural information processing systems 33 (2020): 14350-14361.

---

### Official Review · Reviewer_hJrD · 2022-07-11

**Rating:** 6
**Confidence:** 3
**Soundness:** 3 good
**Presentation:** 2 fair
**Contribution:** 3 good

**Summary:**

The authors propose a hierarchical, non-parametric Bayesian model to identify neural sequences from neural spike data and then to categorize the neural sequences, all in an online manner. Specifically, their model consists of a Dirichlet nonhomogeneous Poisson process (DPP) prior for observed spikes and a Dirichlet Hawkes process (DHP) prior for the neural sequences generating those observed spikes. They refer to their model as the Hierarchical Dirichlet Point model (HDPP). They present a particle filtering method to perform inference under their model as well as a scheme for merging and pruning neural sequence categories/types that may have been incorrectly generated early during inference. They compare their method to two other top-down unsupervised methods, ConvNMF and PP-Seq, on synthetic data. They demonstrate that their method is comparable in performance to PP-Seq under increasing background noise rate, while the performance of ConvNMF steadily drops. They then demonstrate their method on two experimentally-collected neural spike datasets, and present results indicating that their method infers the correct number and types of sequences, converge to similar parameter ranges in independent runs, and a constant expected tune cost with increasing number of observations.

**Questions:**

- How does the time complexity of this method compare to PP-Seq on a fixed-sized dataset (such as the songbird or rat datasets used)? In other words, what is the baseline time complexity cost imposed by the additions in our method (e.g. the non-parametric formulation)?
- What is the dataset size at which using HDPP faster than PP-Seq?
- Figure 5 shows the time cost of the HDPP method, which the authors imply should remain constant as a function of number of detected sequences. What was the number of detected sequences, as a function of the x-axis?
- In Figure 5, why is there an early higher time cost (~50-100s) before settling in?

**Limitations:**

The authors do not mention any limitations of their method nor do they mention potential negative societal impact of their work. Please discuss in greater depth, for example, the tradeoffs that your method's non-parametric formulation and resulting inference algorithm may result in terms of complexity, uncertainty, or power. When, for example, would using PP-Seq be more desirable?

**Strengths And Weaknesses:**

**Strengths**
- Neural recordings for neuroscience are increasingly growing in size and length and under fewer experimental constraints. This direction of neuroscience research begets the need for methods that identify neural sequences in a streaming and non-parametric way. The authors' method would provide an important set of contributions to the field.
- The authors' exposition of their model and inference method is clear.

**Weaknesses**
- The authors should improve the clarity of the conclusions that they draw from their results.
- The authors validate their method against two existing methods, in particularly using PP-Seq to validate the number and type of sequences detected. I recommend that the authors expand the depth and rigor of their evaluations and their analysis of their method. For example, the authors do not demonstrate their method on a large dataset or provide evaluation of time and memory costs of their method vs. another method such as PP-Seq. I would also like to see their method perform on a dataset with a larger number of neural sequences, where there may greater uncertainty in sequence types.

---

> ### Author Response · Authors · 2022-08-02
> **Response**
>
> Thanks for the feedback and constructive suggestions. We have made several improvements according to your questions and comments. Hopefully these will resolve most of your concerns, and that they can be taken into account when deciding the final review score.
>
> The main modifications are as follows:
> * Extra experiments were carried out:
>    * We explored the performance of our method when there are larger types of neural sequences in Figure 2(c).
>    * We tested the time cost and memory cost of our methods vs PP-Seq on the hippocampal data in Figure 4(e) and supplement material Figure 3.
>    * We investigated the time cost as a function of the number of detected sequences in Figure 4(f).
> * We have added the limitation of our approach in the Discussion.
> * We have clarified the conclusions that we draw from the results.
>
> With these modifications, the soundness and presentation of the revised paper has improved.
>
> > For example, the authors do not demonstrate their method on a large dataset or provide evaluation of time and memory costs of their method vs. another method such as PP-Seq.
>
> *  We thank the reviewer for this suggestion, and we have performed the time/memory costs comparison of our method vs parallel PP-Seq on the hippocampal data in Figure 4(e) in the main text and Figure 3 in the supplementary material. To have a fair comparison, we reimplemented our method via Julia, which is the programming language used by PP-Seq. Compared with parallel PP-Seq, our method has much fewer time costs, but higher memory allocations. We believe the cause of such high memory allocation comes from particle resampling, which involves many copy operations to transfer sufficient statistics among particles.
>
> > Figure 5 shows the time cost of the HDPP method, which the authors imply should remain constant as a function of number of detected sequences. What was the number of detected sequences, as a function of the x-axis?
>
> * We thank the reviewer for this useful suggestion, and we have added a new figure about time cost per 3000 spikes as a function of the number of detected sequences in Figure 4(f). The results empirically verify that the time cost per spike tends to be constant in our method.
>
> > In Figure 5, why is there an early higher time cost (~50-100s) before settling in?
>
> * The time cost in Figure 5 is tested via our Python implementation, which is an interpreted programming language and may cause an early higher time cost due to the memory allocations. To have more stable memory management, we retested the time cost via our Julia implementation. The result is presented in Figure 4(e) of the revised paper.
>
> > I would also like to see their method perform on a dataset with a larger number of neural sequences, where there may greater uncertainty in sequence types.
>
> * As suggested, we have added an extra synthetic data experiment in the revised paper. Please refer to Figure 2 and Sec 6.1. This synthetic dataset has 11 different types of sequences with overlapping neurons (6 smaller sequences and 5 larger sequences). We compared our method with PP-Seq. According to the results, our method can identify all types of sequences without prior information on the number of types, while PP-Seq identifies the larger sequences when we set the number of types to 11.
>
> > How does the time complexity of this method compare to PP-Seq on a fixed-sized dataset (such as the songbird or rat datasets used)?
>
> * For both songbird dataset and rat datasets, the time complexity of our method is $O(SP(K+M))$, where $S$ is the number of spikes in these datasets, $P$ is the number of particles, $K$ is the number of inferred sequences, and $M$ is the number of inferred types. As for PP-Seq, its time complexity is $O(NSKM)$, where N is the number of iterations for MCMC sampling, $S$ is the number of spikes, $K$ is the number of inferred sequences, and $M$ is the number of inferred types. Overall, our method’s time complexity is much smaller than PP-Seq since $P << N$.
>
> > What is the dataset size at which using HDPP faster than PP-Seq?
>
> * According to Figure 4(e), our method is faster than parallel PP-Seq on all five hippocampal recordings (3-minutes, 6-minutes, 9-minutes, 12-minutes, and 15-minutes).
>
> > The authors do not mention any limitations of their method nor do they mention potential negative societal impact of their work.
>
> * Thank you for this comment. We have updated the paper to address the technical limitations of our method in Discussion.
>
> > The authors should improve the clarity of the conclusions that they draw from their results.
>
> * We thank the reviewer for this suggestion, and we have revised the paper to clarify our conclusions drawn from the results.
>
> We again thank the reviewer for their comments, constructive feedback, and interesting suggestions, which greatly help improve the quality of our paper. Many thanks.

---

> > ### Comment · Reviewer_hJrD · 2022-08-09
> > **Follow-up question**
> >
> > Thank you for the responses and additional experimentations. I have upgraded my rating from a 4 to 6 because the additional experiments have addressed my questions and concerns.

---

> > > ### Author Response · Authors · 2022-08-09
> > > **Response**
> > >
> > > Many thanks for considering our responses and for revising the score! We will be happy to address any further questions or concerns about the work.

---

### Official Review · Reviewer_dbDh · 2022-07-13

**Rating:** 7
**Confidence:** 3
**Soundness:** 2 fair
**Presentation:** 3 good
**Contribution:** 3 good

**Summary:**

This paper proposes an unsupervised learning approach to detect and cluster neural spike sequences in neural spike data. To model the data, the authors propose a hierarchical Dirichlet point process model, which employs Hawkes processes to model the temporal dynamics of neural activity within sequences, whereas spike rates within a sequence are modelled via a non-homogeneous Poisson process. The authors derive a particle filter based online inference method to detect sequences and infer their types. They use conjugate priors to derive closed-form updates for obtaining posterior distributions of model parameters. The authors evaluate the performance of their method on both synthetic and real datasets.

**Questions:**

- Have you considered or run experiments where you analysed the convergence behaviour of your methodology by assessing how often you manage to recover ground-truth parameters if you optimise on data generated by the process described in section 4, lines  145-152?

- Why have you not compared PP-seq across the board? How does PP-seq's likelihood compare with the likelihood of your approach?

- Can you elaborate what do you compute ROC curves on in section 6.1? Why to you use just one type of sequence for the experiment?

- How does your method perform if there is a greater (spatial and/or temporal) overlap between sequences than what we can see in synthetic datasets?

**Limitations:**

I have highlighted technical limitations and weaknesses above. I have nothing further to add here.

**Strengths And Weaknesses:**

Strengths:
- The authors propose a well-grounded probabilistic generative model for modelling neural activity sequences in spike train data
- The model is able to infer the number and types of sequences from the data
- The authors have devised a tailored particle filter to infer unobserved latent variables for identifying neural activity sequences and inferring their types, which in turn allows for closed-form updates of posterior distributions over model parameters
- The results on artificial data indicate that the proposed methodology can detect and correctly identify the types of underlying neural sequence activity in the presence of background noise.
- The results on real data are qualitatively interesting and comparable to the results of another approach.
- The proposed methodology in principle is scalable to larger datasets with up to 210k spikes.

Weaknesses:
- No analysis is done to show how often the inference procedure leads to suboptimal solutions, which requires reruns of measures like merging and pruning as highlighted by the authors.
- Experiments on synthetic data involve very low number of sequence types (up to 3), so it is hard to see how the (inference) method would perform for a larger number of sequence types
- Comparison with the alternative approach was only done on one data set.

---

> ### Author Response · Authors · 2022-08-02
> **Response**
>
> Thank you for the encouraging feedback and practical suggestions. We have made several improvements according to your questions and comments. Hopefully these will resolve most of your concerns, and that they can be taken into account when deciding the final review score.
>
> The main modifications are as follows:
> * Extra experiments were carried out:
>     * We analyzed the performance of our method when there are larger types of neural sequences in Figure 2(c).
>     * We explored the performance of our method when there is a greater spatial overlap between sequences in supplement material Figure 2.
>     * We analyzed the convergence behaviour via the ability to recover ground-truth parameters in supplement material Figure 4.
>     * We added extra experiments to compare our method with alternative approaches on synthetic data and neural recordings. Please see Figure 2(c), Figure 4(e), and supplement material Figure 1.
>
> With these modifications, the soundness of the revised paper has improved.
>
> > Experiments on synthetic data involve very low number of sequence types (up to 3), so it is hard to see how the (inference) method would perform for a larger number of sequence types.
>
> * We thank the reviewer for this useful suggestion, and we have added an extra synthetic data experiment in the revised paper. Please refer to Figure 2 and Sec 6.1. This synthetic dataset has 11 different types of sequences with overlapping neurons (6 smaller sequences and 5 larger sequences). We compared our method with PP-Seq. According to the results, our method can identify all types of sequences without prior information on the number of types, while PP-Seq identifies the larger sequences when we set the number of types to 11.
>
> > Comparison with the alternative approach was only done on one data set.
>
> * As suggested, we have added extra experiments to compare our method with alternative approaches on synthetic data and neural recordings. Please see Figure 2(c), Figure 4(e), and supplement material Figure 1.
>
> > Why have you not compared PP-seq across the board? How does PP-seq's likelihood compare with the likelihood of your approach?
>
> * We have modified Figure 3 to show the comparison of log-likelihood between our method vs PP-Seq. From the result in Figure 3(e), our method has a similar log-likelihood on the training/testing dataset compared with PP-Seq.
>
> > Can you elaborate what do you compute ROC curves on in section 6.1? Why to you use just one type of sequence for the experiment?
>
> * We follow the steps in Section F.1 of PP-Seq’s supplementary material to compute ROC curves. Specifically, we discretized the sequence times into several time bins. For every time bin, we computed the empirical probability that it contained a sequence from the model, and this resulted in a discretized, nonnegative temporal factor $h$. We then encoded the ground truth sequence times in a binary vector and computed ROC curves by thresholding $h$ over a fine grid of values over the range [0, max($h$)]. Besides, we use this experiment setup (one type of sequence) since it keeps in line with previous literature such as PP-Seq and ConvNMF.
>
> > Have you considered or run experiments where you analysed the convergence behaviour of your methodology by assessing how often you manage to recover ground-truth parameters if you optimise on data generated by the process described in section 4, lines 145-152?
>
> * We thank the reviewer for this practical suggestion, and we have added an extra convergence experiment in the revised paper. Please refer to the supplement material Figure 4. This figure shows the comparison of ground-truth parameters and learned parameters with 95% credible intervals. The results empirically verify that our method could stably recover ground-truth parameters over different runs.
>
> We again thank the reviewer for their comments, constructive feedback, and interesting suggestions, which greatly help improve the quality of our paper. Many thanks.

---

> > ### Comment · Reviewer_dbDh · 2022-08-09
> > **Updating my score**
> >
> > I thank the authors for conducting additional experiments and explanation. I think the paper has overall improved as a result of authors efforts to address raised questions and concerns, I have therefore decided to raise my score.

---

> > > ### Author Response · Authors · 2022-08-10
> > > **Response**
> > >
> > > Thank you very much for considering our responses and for revising the score! We will be happy to address any further questions or concerns about the work.

---

### Meta-Review · Area_Chair_Eg7d · 2022-08-29

**Recommendation:** Accept
**Confidence:** Certain

**Metareview:**

This paper describes a hierarchical Bayesian latent model to identify neural sequences from spike data. Especially in neuroscience, detection of patterns in neural sequences is an important computational problem as the infrared patterns are useful for characterizing brain activity. The key problem is reminiscent of clustering where individual spikes are associated with sequences.

The proposed model -- Hierarchical Dirichlet Point model (HDPP) -- consists of a Dirichlet nonhomogeneous Poisson process (DPP) prior for observed spikes and a Dirichlet Hawkes process (DHP) prior for the neural sequences generating those observed spikes.

Inference is done with sequential Monte Carlo, including a proposal mechanism for merging and pruning neural sequence categories/types that may have been incorrectly generated early during inference. A comparison of the method to two other top-down unsupervised methods (ConvNMF and PP-Seq) on synthetic data is provided.

While the description of the hierarchical model seems to be complete, the reviewers asked for clarifications about the motivations. During the rebuttal, the authors were also able to answer various issues about experimental section and regarding the inference procedure,
They were able to include results of further experiments. As a results, reviewers decided to raise their grades for the paper.

In light of the importance of the problem and the soundness of the methodology, I am inclined to suggest acceptance for this work.


**Award:**

No

---

### Decision · Program_Chairs · 2022-09-14

Accept